# The Influence of a Reflective Identity Leadership Intervention on Perceived Identity Leadership, Social Identity, and Psychological Safety in Cricket

**DOI:** 10.3390/bs14080655

**Published:** 2024-07-29

**Authors:** Adam Hoult, Paul Mansell, Matthew J. Slater

**Affiliations:** Centre for Applied Psychology and Performance, Staffordshire University, Leek Road, Stoke-on-Trent ST4 2DF, UK; a.hoult@lsf.org (A.H.); paul.mansell@staffs.ac.uk (P.M.)

**Keywords:** coaching, teamwork, leadership, cricket

## Abstract

The purpose of this study was to investigate the influence of an identity leadership-framed reflective practice intervention on perceptions of leadership, social identity, and psychological safety in cricket. Building on previous evidence, an eight-week design included three intervention group coaches and their athletes (*n* = 32) and three control group coaches and their athletes (*n* = 34). Measurements of perceived coach identity leadership, social identity, and psychological safety were completed by cricket athletes at week 0 and week 8 for both groups. Intervention group coaches completed three identity leadership-framed reflective tasks in weeks one, three, and five, while the control group coaches continued their regular practices. Controlling for baseline scores, our analysis indicated that compared to the control group, the intervention group athletes reported significantly greater coach identity leadership behaviours, social identity, and psychological safety following the intervention. Social validation data highlighted shared identity, relationships, and learning as potential mechanisms for the positive results seen.

## 1. Introduction

“Cricket is not a sport for everybody” noted the Holding up a Mirror to Cricket report recently published by the Independent Commission for Equity in Cricket (ICEC). As Bishop [1] highlighted in response to this, leadership in cricket needs to care about the values of their organisation and people above any profits or short-term targets. Indeed, the primary role of leadership in any domain is to create an environment of shared belonging to promote performance and wellbeing-related behaviours to intrinsically occur [2]. One way to energise the collective and create a sense of “we-ness” in sports environments is through identity leadership [3]. Speaking to this point, a review by Stevens et al. [4] reported growing evidence that a variety of performance and health-related outcomes (e.g., effort, wellbeing, and performance) are positively influenced by individuals who engage in identity leadership. Leaders who focus on the “must wins” and the “medal count” may be likely to overlook the fundamental human necessity of belonging and may not recognise that performance and psychological well-being are two sides of the same coin. Therefore, developing ways in which sports leaders can enhance their social identity-development skills with their teams is critical. Building on the applied research of Brown and Slater [5], in the current study, we examine the influence of a social identity-informed reflective practice intervention that is tailored to enhance identity leadership behaviours in the novel setting of elite and amateur cricket.

### 1.1. The Social Identity Approach to Leadership

Traditional approaches to leadership have typically focused on the qualities of the leader in isolation, which somewhat overlooks the importance of context and connection to the group [2]. Moving on from the individualistic and traditional approaches to leadership, within identity leadership, scholars have proposed that successful leadership is bound up within group processes [3]. Accordingly, within the identity leadership perspective, it is recognised that optimal leadership behaviours vary according to the content of the group’s identity and context, and at its core, successful leadership involves creating a shared social identity [2]. Social identity can be understood as an individual’s knowledge that they belong to a certain group that has psychological value and significance to them [6], and in cricket, can be encapsulated by the extent to which athletes identify psychologically as a member of their cricket team.

Identity leadership comprises four specific behaviours: (1) defining what the group stands for (i.e., entrepreneurship); (2) embodying the central qualities and attributes of the group (i.e., prototypicality); (3) advancing and standing for the group’s goals and interests (i.e., advancement); and (4) implementing activities, structures, and events that enable members to live out their collective identity (impresarioship) [3,7]. The positive influence of individuals engaging in identity leadership behaviours has been well-documented in the literature. Outside of sport, in a meta-analysis involving 128 studies, Steffens et al. [8] reported a moderate to large positive effect of prototypicality—which is the extent to which a leader is seen to embody shared a social identity—on evaluative and behavioural outcomes such as leadership effectiveness. In sport and exercise, being led by an individual who demonstrates identity leadership behaviours has been shown to improve performance and the maintenance of greater effort [9] and increase group identification and attendance [10]. As well as these motivation and performance-related indicators, researchers have found that when athletes report that their leaders engage in identity leadership, this contributes to positive appraisals of competitive events [11] and greater psychological safety [12].

Overall, the social identity perspective may be well placed to facilitate the overlap between performance and health, and, by engaging in identity leadership, it may prove possible to simultaneously improve both performance and health-related outcomes [4]. Taken together, this evidence suggests that the success of the group may depend on the leader’s ability to improve their interactions and shape a team environment that allows athletes to feel valued, connected, confident, and safe [13]. One concept that speaks to this point and has been examined in the context of identity leadership, i.e., [12], is psychological safety.

### 1.2. Psychological Safety

The primary use of the term psychological safety was derived from the work by Edmondson [14] in the field of organisational psychology. Psychological safety represents a shared belief amongst a group that they are safe to voice ideas and concerns and engage in interpersonal risk taking. Psychologically safe environments increase learning behaviour, performance, communication, innovation, and individual wellbeing [15]. Those environments with lower levels of psychological safety tend to promote fear, conformity, and an accentuated power distance between leaders and followers [16]. Thus, the leader’s role in promoting psychological safety is therefore crucial.

Indeed, Edmondson and Lei [17] noted that leadership is a key antecedent of psychological safety. Research across the organisational context has highlighted the importance of connecting with others [18], engaging in supportive behaviours, and fostering bonds between team members [15]. Nevertheless, some studies have challenged the concept of psychological safety, highlighting its complex nature across different contexts. For example, psychological safety has been found to have an impact on effort levels [19] and the development of unethical behaviours [20]. In a large study consisting of 170,000 teachers, Higgins et al. [21] found that psychological safety was not necessarily supportive of performance over time, and it was more likely that lower levels of psychological safety and higher perceived levels of accountability led to the best performances over time.

Psychological safety is not the absence of being critical but is better reflected in high-quality support through trust and high-quality connection [22]. In a sports team environment, where internal competition is high, an athlete’s ability to have candid conversations and navigate conflict makes it possible to enhance psychological safety [23]. The coach and team environment are therefore crucial to facilitating honest conversations, high levels of team performance, and positive athletic experiences [24]. Having a coach who can reflect on their ability to positively influence the environment is important as sports have been shown to rely on potentially beneficial exposure to environments where it is not entirely safe to fail [25]. 

Furthermore, researchers have provided support for further investigating the link between leadership and psychological safety in sports. Gosai et al. [13] collected data from 166 team sport athletes and found that certain elements of leadership such as consideration, intellectual stimulation, and fostering acceptance of group goals were positively associated with psychological safety and coach–athlete relationship quality. Examining the social identity approach to leadership and psychological safety, Fransen et al. [12] adopted a cross-sectional design with 289 handball athletes. The results indicated that coaches, captains, and informal leaders all had a unique contribution to strengthening team members’ identification with their team. Through this shared sense of ‘us”, in turn, athletes felt psychologically safe in their team to speak up, provide input, and take risks. The study by Fransen et al. [12] has provided important initial evidence of the link between identity leadership and psychological safety that warrants further investigation, and this reflects one area that we sought to build upon. The current study will add further evidence on the associations between identity leadership and psychological safety by assessing the influence of a social identity-informed reflective practice intervention on identity leadership, social identity, and psychological safety. If the intervention influences changes in identity leadership over time, which are also mirrored by changes in psychological safety (i.e., both variables increase from baseline to post-intervention), this would provide further evidence of the positive association between both concepts. 

### 1.3. Reflective Practice

Reflective practice within sports coaching is positioned as an individual meta-cognitive strategy, progressing from simply considering ‘what coaches do’ to also including ‘what, and how coaches think’ [26]. Reflection is an essential part of coach learning [27] but can be challenging and complex due to the wealth of personal experiences that coaches carry [28,29]. To illustrate, with 12 high-level coaches, Nash et al. [30] identified that it is the combination of how to reflect, and upon what criteria, that makes reflective practice a powerful tool to develop expertise. Reflective practice can lack effectiveness if the details regarding what reflection is and how to effectively do it go unstated [26,31]. Despite the appeal of reflective practice, limited empirical evidence exists to support how reflective practice develops effective leadership [32].

Critics of the traditional approach to reflective practice have suggested that it prompts a natural gravitation towards what one lacks, perceiving the coach as a ‘problem fixer’ and ultimately constructing a world where problems become a central focus [33]. Strength-based reflection, which is designed to focus on positive aspects of performance, has been found to create an open-minded and dynamic practitioner [33] as well as increase personal satisfaction and performance [34]. Coach education courses could do more to fully realise the benefits of reflective practice by allowing coaches to develop craft and tacit knowledge, creating flexibility and emotional awareness to respond in the moment [30,33].

Brown and Slater’s [5] novel reflective practice intervention created opportunities for sports coaches to reflect at a team/group-based level. The researchers found that reflecting on coaching practice through a social identity lens elicited significant improvements in athletes’ rating of their coaches’ engagement in the identity leadership behaviours of advancement, entrepreneurship, and impresarioship. Providing a focus for coach reflective practice is a useful way for coach education to go beyond the technical and tactical and move towards developing a sense of “we” and “us” within each context. That said, Brown and Slater [5] did not measure additional variables beyond identity leadership, and so the influence of the social identity-informed reflective practice intervention on pertinent outcomes such as social identity and psychological safety remains unknown. 

### 1.4. Leadership Development Interventions

Despite the growing extant literature on the consequences of identity leadership, social identity, and psychological safety individually and collectively, there remain comparatively few applied interventions and studies to assess the efficacy and utility of developing these key behaviours and concepts. Within the social identity paradigm, researchers have aimed to take theory to practice through the creation of identity leadership interventions such as the 3Rs [35] and the 5Rs [36].

The 3R model involves working with leaders and their teams through the phases of Reflecting, Representing, and Realising to enhance their identity leadership skills and behaviours, and ultimately, create a shared sense of social identity [2]. In one study over a two-year period, implementing the 3R model on two occasions in an international disability sports team, Slater and Barker [35] found that, compared to baseline data, an athlete’s social identity and the number of hours practised away from formal camps increased as a result of the intervention. Despite the elite context, which can be seen as a strength, the research was conducted with a single team each year, and this resulted in a small sample size. In addition, using the 5R framework, Mertens et al. [37] addressed the small sample size and moved knowledge closer to cause and effect by conducting an experimental test of the effectiveness of 5R^s^ that was found to strengthen athletes’ identification with their team. There were some limitations that can affect the conclusions made from this study. Two limitations that Mertens et al. [38] set about addressing were the use of a waitlist control group for all coaches who were willing to take part in the study and the inclusion of female teams. The results from Mertens et al. [38] indicated that the 5R shared leadership program was useful for developing high-quality leadership, nurturing player’s health, and improving overall team functioning. Haslam et al. [39] measured additional variables within their 5R study, most notably teamfullness (of which one element was psychological safety). The results indicated that leaders, relative to the control condition, reported medium-sized increases in team engagement and teamfullness. Overall, these studies highlight the benefits of collaborating with the leaders and their teams, and that developing one’s identity leadership could provide the tools to enhance leadership success.

### 1.5. The Current Study

Building on the preceding backdrop and the identity leadership-framed reflective practice intervention developed by Brown and Slater [5], we aim to apply the same intervention within cricket specifically, in male and female teams, and with the addition of social identity and psychological safety as novel variables. There is a clear need to build on this work as identity leadership and psychological safety may be aspects that contribute to effective environments that can be used as a springboard to flourish in sports [13]. We hypothesised that compared to the control group, the identity leadership-framed reflective practice intervention will elicit a significant increase in athletes’ perceptions of their coach engaging in identity leadership behaviours (H1), social identity (H2), and psychological safety (H3) from baseline to post-intervention. 

## 2. Method

### 2.1. Participants and Design

We adopted a 2 (group: experimental vs. control) × 2 (time: pre vs. post) design. Using a medium to large effect size (*d* = 0.66) reported in Brown and Slater [5], we conducted apriori power analysis using G*Power 3.1.9.7. We set an alpha of *p* = 0.05 and a power of 0.8, resulting in a required sample size of 67. Due to the typical sizes of cricket teams and anticipated retention rates, we aimed to recruit six U.K. coaches. A total of 6 coaches and 85 of their athletes were recruited for the study, consisting of (1) 2 ‘competitive elite’ adult female teams [40]; (2) 2 adult amateur male teams that competed at division one or above in their county competition; and (3) 2 independent school male first teams, which competed at the U18 level. One team from each of these three clusters was randomly allocated to the intervention group and one to the control group. Our retention rate was 77.6%, resulting in a final sample of 66 athletes (intervention *n* = 32, control *n* = 34). 

### 2.2. Measures

#### 2.2.1. Identity Leadership

The Identity Leadership Inventory—Short Form (ILI-SF) [7] was completed by athletes to assess perceptions of their coach’s identity leadership behaviours using a scale from 1—strongly disagree, to 7—strongly agree (e.g., “The leader acts as a champion for the group”). Each item assesses one identity leadership behaviour, and the scores across the four items are averaged to produce a single mean score for identity leadership [7]. Results from a range of studies show that the ILI and ILI-SF have good construct and discriminant validity globally, with the short form well-suited to repeated measurement, which is inherent in intervention research [7,41]. 

#### 2.2.2. Social Identity

The athletes completed the Social Identity Questionnaire for Sport (SIQS) [42]. The SIQS is a 9-item, positively worded questionnaire and athletes answered the questions in reference to their cricket team using a 7-point scale from 1—strongly disagree to 7—strongly agree (e.g., “I feel strong ties to other members of this team”). The SIQS can be used to assess the three aspects of social identity: (1) in-group ties; (2) cognitive centrality; and (3) in-group affect, or as an overall score for social identity. In this research, the SIQS was used as a global construct to analyse the athletes’ general strength of social identity in relation to their cricket team. Support for measuring social identity as a global construct is provided [42] and used in Mertens et al. [38]. 

#### 2.2.3. Psychological Safety

The Team Psychological Safety Questionnaire [14] was used to assess the athlete’s perceptions of psychological safety within the athletes’ team. Players completed a 7-point scale from 1—strongly disagree to 7—strongly agree (e.g., “It is safe to take a risk on this team”). The measure has been used extensively across other contexts and by Fransen et al. [12] in sports. Fransen et al. [12] investigated the reliability and structural validity during their cross-sectional study and found that after exploratory factor analysis, the removal of item 6 provided greater validity and reliability as opposed to the original 7-item questionnaire. Although it is possible to argue that some items may not be as relevant in the sporting context as they are in other team settings, the measure is popular within the literature, and as there is limited research in the sporting context, and in accordance with Fransen et al. [12], the adapted 6-item measure was used in this study.

#### 2.2.4. Social Validation

A social validation questionnaire was used to investigate the effectiveness of the overall intervention process and was given to all participants within the intervention group [43]. Adapting previous examples [44], the questionnaire featured eight questions, which were measured on a 7-point Likert scale from 1—do not agree at all to 7—agree completely (e.g., “The coach is strongly motivated to engage with this team”), and eight open-ended questions (e.g., “Please describe whether the coach helps you to feel as though you belong to the team?”). Drawing on the principles of reflexive thematic analysis [45], the data were analysed guided by six phases, which included (1) familiarisation of data; (2) initial codes: (3) creation of themes; (4) reviewing themes; (5) defining theme names; and (6) producing the report, which can be found in the Results section. The qualitative data collected via social validation were based on Brown and Slater [5] and the purpose was to complement the quantitative data gathered. We sought to optimise the quality, trustworthiness, and rigour of the qualitative data [46]. The data were gathered and analysed by the first author, and they shared their analyses with the second and third authors in August and September 2023 to debate, discuss, and act as critical friends. This critical friend process was not to verify the data but rather to question and explore the generated codes and themes in the context of participant data/lead researcher experiences. This journey continued during the write-up phase of the research paper. Further, the lead researcher reflected on his own biases and experiences during the journey of data analysis and write-up. 

### 2.3. Procedure

Initial information was sent to pathway development offices, technical directors, and head coaches of cricket teams along with invitations to engage in an initial phone call to discuss the nature of the research. Interested coaches then received further information and consent details if they were willing to participate. After completing this, players were given information regarding the study, parental information and consent (U18 teams), and informed consent. The athletes who wished to participate completed the baseline measures using an online link on personal devices or using paper copies, which were then sealed and taken by the lead researcher (first author). Following the collection of baseline data (i.e., week 0), the two coaches within each cluster—(1) adult male, (2) adult female, and (3) U18—were randomly assigned to the intervention group or the control group. All players completed the measures at baseline and post-intervention (i.e., week 8), with the intervention group only completing social validation post-intervention. At both the baseline and post-intervention sessions, athletes were instructed to read all of the information, that participation was voluntary, to seek clarification regarding any confusing terms or questions, that their identity would remain anonymous throughout the process, and that they could withdraw.

### 2.4. The Intervention

Following the collection of baseline data (week 0), the three coaches in the intervention group received task one in week one via email. The coaches had seven days to complete this activity, which was directly emailed back to the lead researcher. The coaches subsequently completed task two in week three and task three in week five. The identity leadership-informed reflective tasks were in line with the intervention used in Brown and Slater’s [5] study. Task one was a combination of strength-based reflection and the four principles of identity leadership [3,34] with coaches required to answer questions such as “what do I want to start doing?” and “what do I want to do more of?”. Task two combined a targeted reflective narrative with identity leadership behaviours [47,48] that required coaches to identify a specific instance in their coaching that they believed highlighted identity leadership components using the format of identification, description, significance, and implications [49]. Coaches were asked to identify a situation that fitted one or more of the identity leadership behaviours, followed by a rich description of thoughts, feelings, and circumstances experienced. Task three combined a reframing exercise through an identity leadership lens [50,51], which required each coach to identify a new occurrence and consider the perspective of someone else in the group. Coaches would then reflect and comment on their own actions in relation to each of the four identity leadership behaviours. Tasks were completed digitally in the coach’s own time across seven days, which was then emailed back to the lead researcher. The procedural elements of the intervention are shown in Table 1. All tasks are available from the first author on request.

## 3. Results

To assess H1, H2, and H3, we conducted a one-way ANCOVA (group: intervention vs. control) to examine post-intervention differences in perceived coach identity leadership, social identity, and psychological safety, controlling for baseline scores. We adopted ANCOVA as this covariance approach is the optimum statistical estimation in baseline to post-intervention randomized control trial designs, particularly—as we found in the current study—when baseline scores for the intervention and control group are different [52]. In other words, while there is debate surrounding whether a 2 × 2 mixed model or ANCOVA is best practice in designs such as ours, we deemed ANCOVA as the best approach for our data because the intervention and control group reported significantly (all *p*’s < 0.001) different baseline scores across all variables. Adjusted Bonferroni post-hoc pairwise comparisons were planned to follow up on any significant result. See Table 2 for descriptive statistics and effect sizes. 

### 3.1. Identity Leadership

Consistent with H1, a one-way ANCOVA (group: intervention vs. control) indicated a significant difference between the groups on identity leadership, *F* (1,63) = 44.15, *p* < 0.001, *η*^2^*_p_* = 0.412. Controlling for baseline scores, at post-intervention, the intervention group (*M* = 5.37, *SD* = 0.77) reported greater identity leadership compared to the control group (*M* = 4.94, *SD* = 0.71). Compared to baseline data, the increase in the intervention group reflected a large practical improvement (*d* = 1.20) in identity leadership.

### 3.2. Social Identity

Consistent with H2, a one-way ANCOVA (group: intervention vs. control) indicated a significant difference between the groups on social identity, *F* (1,63) = 20.96, *p* < 0.001, *η*^2^*_p_* = 0.250. Controlling for baseline scores, at post-intervention, the intervention group (*M* = 5.06, *SD* = 0.67) reported greater social identity compared to the control group (*M* = 4.88, *SD* = 0.61). Compared to baseline data, the increase in the intervention group reflected a large practical improvement (*d* = 0.90) in social identity.

### 3.3. Psychological Safety

Consistent with H3, a one-way ANCOVA (group: intervention vs. control) indicated a significant difference between the groups on psychological safety, *F* (1,63) = 34.35, *p* < 0.001, *η*^2^*_p_* = 0.353. Controlling for baseline scores, at post-intervention, the intervention group (*M* = 5.06, *SD* = 0.67) reported greater psychological safety compared to the control group (*M* = 4.88, *SD* = 0.61). Compared to baseline data, the increase in the intervention group reflected a large practical improvement (*d* = 0.92) in psychological safety.

In summary, the results indicated that when controlling for baseline scores, at post-intervention, the intervention group reported significantly higher levels of identity leadership, social identity, and psychological safety compared to the control. Accordingly, the intervention brought about large practical (effect size range: *d* = 0.90–1.20) increases in how the athletes reported their coach’s identity leadership and their own feelings of social identity and psychological safety in their cricket team.

### 3.4. Social Validation 

Thirty-two participants responded to all of the questions on a 7-point Likert scale regarding the effectiveness of the intervention. Percentages revealed that these participants responded with somewhat agree, agree, and strongly agree. Statements included important to create a shared sense of identity (93.75%), bringing people closer together is important (100%), the coach has a positive influence on this team (93.75%), the coach is strongly motivated to engage with this team (96.87%), the coach can bring the team together (90.62%), I can contribute my ideas to the team (90.62%), the coach values my contribution to the team (90.62%), and I feel a sense of belonging with this team (96.87%). 

Three higher-order themes and associated sub-themes were created (see Figure 1): (1) relationships, (2) learning, and (3) shared identity. The first theme, relationships, was influenced by the coach, specifically by getting to know the players: “he has tried to get to know me better this year and has taken more of an interest in how I view the game”; using positive affirmations: “he often checks in about my life and also about areas of my game that I am really good at”; and through attentive listening: “he has made a big effort to listen to the guys more and not be so uptight about his own philosophy” and “he clearly values our input and always keen to let us have our say”.

The second theme was that of learning in relation to allowing the players to take ownership: “giving us the chance to set the goals for the season was a good way to get us all together and buying in”; providing feedback to players: “he provides us with clear feedback to strive to get better” and “the coaching staff do push me to become better, making sure that I improve every time I leave and make sure I’m learning more about my game”; and ensuring that players have role clarity: “we have a role all of the time, even in training, our roles are clear, and our intentions are clear”.

The third theme was that of shared identity through the connection with all players “he communicates with all of us on an individual level and I feel that we all feel like one connected group”; enthusiasm for the development of the group: “he does not seem to favour anybody and I think there is a real fairness and eagerness to see everybody in the team improve” and “ideas are respected and willing to be listened to”; and the creation of a playing identity: ‘he has allowed us to create a style of play that we think will be the most enjoyable for us to play this season” and “everything is driven by the way we want to play our cricket”. 

The final sub-theme that is important to include here is the lack of voice through relationships with others in the team: “sometimes, it depends which group I am assigned to when planning and post-match chats” and “I do feel safe but only to certain people within the team”; and not fully believing in the team’s direction: “I think we also need to talk about plan b and c. I feel like if I mention this I will be seen as negative and not ‘fitting in’ to the team’s style of play”. These valuable points highlight that even in an environment that has been measured as psychologically safe, there are still individuals who feel that they are unable to speak up. 

## 4. Discussion

The purpose of our study was to examine the influence of an identity leadership-framed reflective practice intervention—as developed by Brown and Slater [5]—on identity leadership, social identity, and psychological safety in cricket. In other words, we aimed to explore if engaging in tailored reflective activities through a social identity lens could lead to increased perceptions of coach identity leadership and, in turn, strengthen bonds between team members and create an environment where all cricketers can feel safe to be their authentic selves. In line with our hypotheses (H1–H3), the results indicated that when controlling for baseline scores, the intervention group reported significantly higher levels of identity leadership, social identity, and psychological safety compared to the control group following the intervention. Further, in terms of the practical meaningfulness of the data, the intervention produced large increases in how the athletes perceived their coach’s identity leadership, the athletes’ strength of social identity, and their psychological safety.

This current research contributes to our understanding in at least three ways. First, we evidence that applying an identity leadership-informed reflective practice intervention directly and independently with cricket coaches can lead to athletes’ enhanced perceptions of leadership. Second, as well as leadership, the intervention positively influenced key performance and health-related variables of social identity and psychological safety, and further documents the positive association between identity leadership and psychological safety. Third, we outline potential mechanisms of change for the intervention and evidence that the reflective intervention has utility in cricket, in elite and amateur settings, and in male and female teams. We structure the following discussion based on these three contributions. 

Within the social identity paradigm, there is growing attention being paid to leadership development programmes including the 3Rs [35] and the 5Rs [36,38]. The findings from the current study add further support for the social identity approach in the development of leadership programmes in sports, and, specifically on the influence of collaborating with sports coaches directly and independently. Both the 3R and 5R programmes typically involve larger-scale programmes of work including a range of activities with leaders and, often all, athletes. In our study, as with Brown and Slater [5], the intervention was delivered to the head coach independently, and then they were tasked with applying their reflections in practice. Accordingly, the head coach is at the core of the influence and utility of the intervention, and there is clear scope for this programme to be delivered as part of coach education pathways in personal development. Moreover, the identity leadership-informed reflective practice intervention in the current study has brought about comparable, and arguably larger (in effect size terms), improvements compared to more intensive and larger-scale 3R and 5R programmes. The use of targeted reflections with the head coach aimed to initiate an identity leadership-based, contextually focused, reflective process to bring about changes in behaviour. In turn, this approach may have provided deeper and more focused grounds for reflection on, in, and for coaching action [32]. Our findings indicated that the coaches were able to action these changes with their teams given that their athletes, without knowing the specific details of the intervention, reported greater levels of identity leadership post-intervention. 

Previous researchers have found that the identity leadership-informed reflective practice intervention improved three of the four identity leadership behaviours [5]. We enhanced our understanding of the positive influence of the reflective intervention by documenting that, compared to a matched control group, the tasks engaged in by the coaches led to the strengthening of cricket athletes’ social identity and psychological safety (as well as identity leadership). In other words, in addition to leadership, there are downstream effects of the identity leadership-informed reflective practice intervention on coaches and athletes over an eight-week period. Furthermore, the positive influence of the intervention on both identity leadership and psychological safety (albeit at a smaller magnitude for psychological safety) provides evidence that a positive large practical shift in identity leadership produced a similar large positive shift in psychological safety. This provides further support for previous evidence that has indicated that identity leadership and psychological safety are positively related [12].

Figure 1 highlights some potential mechanisms for the positive influence of the intervention. These reflect initial and potential mechanisms at play. It can be suggested that creating a collective vision, as part of the key theme of shared identity, is an example of entrepreneurship, which is an act that seeks to unite rather than divide the group [3]. Participant responses on the social validation data highlighted that having a set of guiding principles could create certainty amongst the group whereby all athletes are clear on the team’s direction and their role. Certainty is noted as a key boundary condition and an essential ingredient that enables psychological safety to contribute to performance and learning [17]. Shared leadership could have also been a vital part of this process. By allowing players to take ownership and contribute to the collective vision, feelings of connectedness could have been strengthened along with enhanced motivation to speak up and offer knowledge. This may explain the improvements we document, and it builds on the work by Fransen et al. [12] who found that identity leadership displayed by coaches, captains, and informal leaders all contributed to strengthening social identity and a sense of psychological safety. In other words, allowing players to contribute to the collective vision may have enhanced perceptions of agency and contribution to team processes, thus enhancing psychological safety. Haslam et al. [39] found that leaders who gave team members an opportunity to contribute to its direction reported increases in teamfullness, which consisted of psychological safety, team goal clarity, and an inclusive team climate. Inclusiveness has been shown to enable performance [53] and positively impact psychological safety in health care [54] and is noted as a key strategy when developing high-quality youth sporting experiences [55]. Indeed, creating a collective vision and a sense of “we” and “us” may enhance inclusiveness and allow group members to live out their shared sense of social identity. 

Another potential reason for the positive findings in the current study could be due to collective goal-setting and collaborative feedback. Athlete responses in the social validation data spoke of the benefit of the use of collective goal setting, providing regular opportunities to discuss progress and facilitate feedback. Providing structures and activities for group members to live out their shared sense of social identity is termed impresarioship [3]. Allowing players to provide ideas and contribute to team goals could have created acceptance amongst the group, and it is the fostering of acceptance of these collective goals that can contribute to psychological safety [13]. The results in this study suggest that participants were positively engaging in the feedback process, and if this feedback is aligned with group goals, then the coach’s intentions are clear. The understanding of intentions is crucial when giving and receiving honest feedback [16,17]. Cricket coaches in this study may have opened dialogue with their athletes and allowed them to engage with the feedback process, thus developing interpersonal skills. Understanding the role of feedback to different individuals, creating reciprocity, nurturing the feedback relationship, and having role clarity were all mentioned as key components of coaches who are willing to show that they care [22]. Perhaps through identity leadership, cricket coaches had increased awareness in the moment and thus demonstrated care for group members. 

This ability to display and encourage openness is also critical to the coach–athlete relationship and has been found to positively influence psychological safety [23]. Providing clear roles is an example of impresarioship, and having roles can lead players to internalise a sense of shared group membership, which has been shown to be a basis for in-role performance and organisational citizenship behaviour [41]. Role clarity could have played a part in positive social identity results, but it has also been shown to be a key antecedent of psychological safety [56]. An example from player responses includes the sharing of personal cricket journeys, which is an example of personal disclosure mutual sharing (PDMS). Examples from cricket highlight the importance of this activity, which has been found to develop social identity, friendship identity, and collective efficacy [57], and this study could have contributed to the positive results. It could be argued that PDMS can also foster psychological safety through the mechanisms of vulnerability and comradeship by demonstrating emotions, taking risks to share their journey, and accepting their own and others’ struggles [58]. Examples of prototypicality can also be highlighted in this study. As a result of the reflective task, one of the coaches planned a number of presentations to athletes to be delivered by past players and members of the cricket club’s community. Leaders who are seen to embed themselves into the community could enable the perception that they are embodying the attributes that define what it means to be a member of this group and, in turn, display prototypicality [3]. These examples of impresarioship and prototypicality suggest that opportunities have been created by the leader for athletes to communicate freely, propose new ideas, and voice their feelings to promote the idea that coaching is about people sharing knowledge and constantly learning from each other [23]. 

This said, the sub-theme ‘lack of voice’ highlighted that, even in a team that has made significant improvements in psychological safety, there is still potential for others to feel a lack of direction within the team. This could be due to other factors such as personal performances and organisational fit [59], supporting the notion that psychological safety is dynamic, fragile, and reliant upon a variety of contextual factors [25]. Members of the team may feel psychologically safe when interacting with the coach, but because of the construct’s dynamic nature, this may change depending on other interactions. If the environments were truly psychologically safe, then all members would be able to offer constructive disagreements, regardless of which members of the group they are working with [12]. Even though our results highlight significant improvements in psychological safety, it is perhaps too simplistic to categorise the holistic environment due to the highly variable individual responses [25].

### 4.1. Limitations and Future Directions

Our study has a number of limitations. First, there was no performance measurement included. Psychological safety has been shown to play a critical role in workforce performance [18], and using objective measures such as performance metrics is a crucial step (e.g., runs scored, wickets taken), but there are limited examples of this in sports. Future research may wish to examine the influence of the intervention on performance markers. Second, we adopted social validation as a measurement to add insight from an athlete’s perspective, but we did not glean the coach’s experiences of the intervention quantitatively or via social validation. As a form of triangulation, it could be argued that information gathered from coaches in this way could provide invaluable insight regarding the effectiveness and satisfaction of the intervention and highlight methods employed to bring about change [43]. Further, this research project was conducted in the UK, and there remain opportunities to enrich our understanding of cultural perspectives [60] within the application of identity leadership in sports. There is a need to conduct research with under-represented populations—not solely in cricket—but in various sports systems in the UK and across the globe. There are many potential intersections and opportunities to explore the application of leadership development within sport, and we hope that is embraced by future researchers and practitioners.

As noted by Brown and Slater [5], this eight-week intervention is relatively short-term. Due to the potential fluctuation of athlete perceptions and, perhaps, performance slumps, a more detailed, longer study that includes a follow-up measurement phase (i.e., 3 or 6 months following the intervention) would be valuable to understand the longevity of the positive influence of the intervention. Further, although we gathered the tasks completed by the coaches and the athletes’ perceptions, we did not attend training or competitions. Therefore, beyond the athlete’s social validation data, we do not have an insight into what precisely the coaches altered in terms of their behaviours. In the future, researchers may wish to consider conducting observations. Observations have the potential to shine a light on the challenges that coaches, athletes, and teams face throughout the course of a season. Capturing in situ observations of critical incidents of identity leadership across the season would limit the constraint of relying on participant recall alone and might result in a more comprehensive understanding of cricket specifically [61]. 

### 4.2. Conclusions

The present study has shown that an identity leadership-framed reflective practice intervention, working with cricket coaches directly, can simultaneously develop coaches’ identity leadership skills, athletes’ identification with their teams, and the psychological safety of the group. Circling back to the ICEC report with which we started this article, in the pursuit of making cricket a game for everyone, targeted reflective practice through an identity leadership lens may assist cricket coaches in developing their identity leadership skills, and, crucially, be the creators of a caring and psychologically safe environment. 

## Figures and Tables

**Figure 1 behavsci-14-00655-f001:**
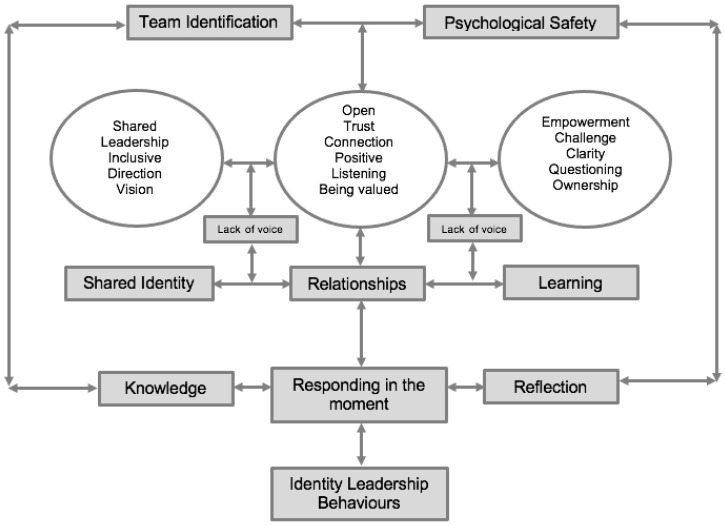
Visual representation of themes developed from responses to social validation results.

**Table 1 behavsci-14-00655-t001:** Overview of the intervention procedure including activities, measures, and consent.

Who	Recruitment	Information and Consent	Week 0	Week 1	Week 3	Week 5	Week 8
Coaches	Initial contact and research information.			Intervention Group: - Identity leadership discussion with coach. - Reflection Task 1: Four key principles and a strengths-based reflection.	Intervention Group: - Reflection Task 2: Social identity-based reflective narrative.	Intervention Group: - Reflection Task 3: Reframing reflection.	Follow up email to thank coaches for their time and effort.
Athletes		Participant information and informed consent.	Baseline Measures Intervention Group: - Identity leadership - Social identity - Psychological safety - Social Validation Control Group: - Identity leadership - Social identity - Psychological safety	Continued training and matches as normal			Post Measures Intervention Group: - Identity leadership - Social identity - Psychological safety - Social Validation Control Group: - Identity leadership - Social identity - Psychological safety.
Parents (U18)		Participant information and informed consent.					Follow up email to thank parents and participants for their involvement.

**Table 2 behavsci-14-00655-t002:** Descriptive statistics and effect sizes (Cohen’s *d*) for identity leadership, social identity, and psychological safety across the control group and SIL intervention group.

Variable	Control Baseline Mean (*SD*)	Control Post Mean (*SD*)	Effect Size (*d*)	Intervention Baseline Mean (*SD*)	Intervention Post Mean (*SD*)	Effect Size (*d*)
Identity Leadership	4.87 (0.82)	4.94 (0.71)	0.09	4.12 (1.04)	5.37 (0.77)	1.20
Social Identity	4.97 (0.61)	4.88 (0.61)	−0.15	4.01 (1.17)	5.06 (0.67)	0.90
Psychological Safety	4.83 (0.88)	4.86 (0.77)	0.03	3.89 (1.42)	5.18 (0.82)	0.92

## Data Availability

The raw data can be found in the attached Appendix A (excel format).

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
