# Peer review of "The Influence of a Reflective Identity Leadership Intervention on Perceived Identity Leadership, Social Identity, and Psychological Safety in Cricket"

_behavsci, 2024, doi:10.3390/bs14080655_

Round 1

Reviewer 1 Report

Comments and Suggestions for Authors

I applaud the authors for taking on this important and enlightening study. I believe the results are important and show the impact of coach reflection on athlete experiences. This paper is very well written with only a few minor suggestions made throughout by myself. 

My main area of concern is the results section. There should be 

1. A stronger argument for the use of an ANOCOVA over Repeated Measures or a repeated measures (with a covariate) should have been used. 

2. The qualitative data needs to be better explained in its methodology including the steps of the RTA and a section on trustworthiness as well as reflection by the authors on  their biases brought to the qualitative coding. 

3. I would encourage the addition of assessment of the coaches self perceptions of leadership changes being done in future research. 

Reviewer 2 Report

Comments and Suggestions for Authors

Hi all

Really amazing work from the authors, especially with the design used. 

Some comments to improve it

2.1- Include level a bit more succinctly and align to Swann et al., 2016 taxonomy

2.4- please provide more details on the procedural elements on the intervention, and its mechanisms of it, perhaps in a table to allow replication and extension through future research

Discussion

Since it is about identity leadership, and historically cricket in the UK is a multicultural, multiethnic and multirace sport, I would strongly encourage discussions on cross-cultural sport psychology components which would strengthen this work. Some references below:

Schinke, R. J., Blodgett, A. T., Ryba, T. V., & Middleton, T. R. (2019). Cultural sport psychology as a pathway to advances in identity and settlement research to practice. Psychology of Sport and Exercise42, 58-65.

Gupta, S. (2022). Reflect in and speak out: An autoethnographic study on race and the embedded sport psychology practitioner. Case Studies in Sport and Exercise Psychology6(S1), S1-10.

Burdsey, D. (2011). That joke isn’t funny anymore: Racial microaggressions, color-blind ideology and the mitigation of racism in English men’s first-class cricket. Sociology of sport journal28(3), 261-283.
